# But Are You Sure? An Uncertainty-Aware Perspective on Explainable AI

**Charlie Marx**[*]
Stanford University
ctmarx@stanford.edu

**Youngsuk Park**[†]
AWS AI Labs
pyoungsu@amazon.com

**Hilaf Hasson**
AWS AI Labs
hashilaf@amazon.com

**Yuyang Wang**
AWS AI Labs
yuyawang@amazon.com

**Stefano Ermon**
AWS
ermons@amazon.com

**Jun Huan**
AWS AI Labs
lukehuan@amazon.com

## Abstract

Even when a black-box model makes accurate predictions (e.g., whether it will rain tomorrow), it is difficult to extract principles from the model that improve human understanding (e.g., what set of atmospheric conditions best predict rainfall). The field of model explainability approaches this problem by identifying salient aspects of the model, such as data features to which the model is most sensitive. However, these methods can be unstable and inconsistent, leading to unreliable insights. Specifically, when there are many near-optimal models, there is no guarantee that a single explanation for a best-fitted model will agree with "true explanation": the explanation from the (unknown) true model that generated the data. In this work, we aim to construct an uncertainty set that is guaranteed to include the true explanation with high probability. We develop methods to compute such a set in both frequentist and Bayesian settings. Through synthetic experiments, we demonstrate that our uncertainty sets have high fidelity to the explanations of the true model. Real-world experiments confirm the effectiveness of our approach.

## 1 Introduction

Data is now collected at a much faster rate than can be processed directly by humans. Thus, machine learning has been used to synthesize complex datasets into predictive models. For example, models can predict the 3D structure of proteins from their amino acid sequences [1] and forecast supply chain demand [2, 3]. However, modern models are often black-box in nature, meaning that even when they make accurate predictions, it is difficult to extract interpretable principles or intuitions. Whereas human experts can communicate their reasoning, predictive models typically lack the ability to communicate principles.

In response to this challenge, there has been growing interest in model *explanations*: human-interpretable descriptions of model predictions [4–7]. The explanations highlight aspects of the model that are particularly relevant for some downstream goal, such as calibrating trust in a model or identifying patterns in complex data. Popular explanations include SHAP [8], LIME [9], integrated gradients [10], TCAV [11], and counterfactual explanations [12].

Use cases for model explanations can be organized around two goals: model auditing and scientific inquiry. In model auditing, the goal is to validate or debug the predictions of a trained model. For

---

[*]Work done during an internship at AWS AI Labs.

[†]Correspondence to Youngsuk Park, pyoungsu@amazon.com.

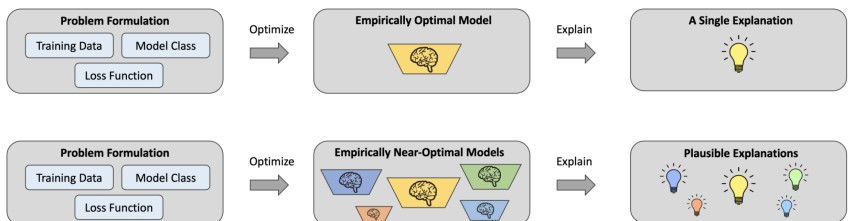

Figure 1: **Top:** The standard explainability pipeline. A single "best-fitting" model is trained and a single explanation is generated from this model. **Bottom:** An instantiation of our proposed explainability pipeline. We explore the set of explanations associated with near-optimal models to construct a confidence set for the explanation of the true model.

example, we might ask *"In what way does this climate model for global surface temperature depend on $CO_2$ emissions?"*. In contrast, in scientific inquiry the object of interest is the data generating distribution itself. An analogous question for scientific inquiry would be *"In what way is the global surface temperature explained by $CO_2$ emissions?"* Explanations used for model audit give insights *about the model*, while explanations for scientific inquiry give insights *about the world*. See Section 7 for further discussion.

Explanations are already being used for scientific inquiry in many domains, such as materials discovery [13], genomics [14, 15], motor vehicle collisions [16], environmental science [17], and finance [18, 19]. Usually, a practitioner chooses a single "best-fitting" model and treats explanations of that model as representative of the data generating distribution. However, model explanations are known to be unstable (i.e., sensitive to small perturbations in the data) [20–25] and inconsistent (i.e., random variations in training algorithms can lead models trained on the same data to give different explanations) [26]. The problem is worsened by the phenomenon of *model multiplicity*: the existence of distinct models with comparable performance [27–29]. If there exist competing models—each of which provides a different explanation of the data-generating distribution [30]— how can we tell which explanation is correct? These issues threaten the applicability of existing explainability procedures for scientific inquiry. Given that explanations are known to vary widely among even near-optimal models [31], we cannot assume an explanation from a model with good performance is representative of the data generating distribution.

In this work, we aim to develop simple and broadly-applicable procedures to use explanations for valid scientific inquiry. Instead of computing the explanation for a single best-fitting model, we wish to infer the explanations of the "true model", or "data generating distribution", of the conditional distribution of the target given the input. To that end, we provide an *uncertainty set* containing plausible explanations for the (unknown) data generating distribution. See Figure 1 for the high-level idea. For a simple illustration in Figure 2, a naive explanation from multiple trained model (or any single explanation within that interval) consistently disagree with the true explanation of a well-specified linear model. In contrast, our principled uncertainty sets include the true explanation.

Our main contributions include:

- We present a framework for explaining the data generating distribution, as opposed to a trained model. We give a simple example where existing explainability procedures fail to recover the true explanation in this framework.

- We propose three simple yet rigorous methods to construct uncertainty sets for true explanations: one for a frequentist setting, and two for a Bayesian setting under different assumptions. We provide finite-sample coverage guarantees for the uncertainty sets given by each method.

- Through simulations, we demonstrate the effectiveness of our method in terms of the coverage, i.e., how often the uncertainty set includes the true explanation, and the size of the uncertainty set. We also apply our methods to real datasets to infer feature importance.

The rest of the paper is organized as follows. After reviewing related works (Section 2), we introduce a framework for quantifying uncertainty in model explanations (Section 3). We then develop frequentist and Bayesian approaches to construct principled uncertainty sets (Sections 4 and 5, re-

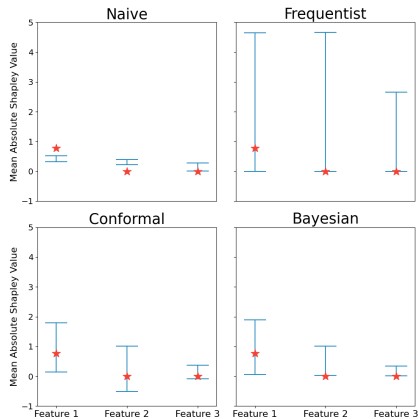

Figure 2: A comparison of methods for computing uncertainty sets for explanations. The explanation is the mean absolute Shapley value, a measure of feature importance. In each panel, the feature importance for the true model that generated the data is marked by a red star. The three features follow a multivariate Gaussian distribution and the first two features are highly correlated. The true labels were sampled from the linear model $y^{(i)} = [1, 0, 0] \cdot x^{(i)} + \epsilon^{(i)}$, where $\epsilon^{(i)} \overset{iid}{\sim} \mathcal{N}(0, 1)$. In the top left panel, we subsample the dataset 100 times and explain a best-fitting linear model for each dataset. Note that the best-fitting model consistently underestimates the importance of Feature 1. In the other three panels, we display confidence intervals generated by the three methods we propose. The frequentist intervals come with strong guarantees, but tend to be wider. The conformal and Bayesian approaches take advantage of additional information (e.g., prior and posterior distribution) to get tighter intervals.

spectively). Finally, we conduct an experimental study on both synthetic and real-world datasets (Section 6), followed by a final discussion (Section 7).

## 2 Related Work

**Explainable AI.** Explainable AI (XAI) aims to present model behavior in a way that humans can easily understand. Some models are inherently more interpretable, such as generalized linear models (GLMs) [32] and tree-based models [33]. For less interpretable models, post-hoc explanations can still provide insights. Popular methods include Shapley-value based approaches [8, 34–36], perturbation-based approaches [37], local approximations [9], tree-based methods [38], and DeepLIFT [39]. Recently, there have been attention to improving the robustness of explanations to distribution shifts [25, 40]. Separately, for probabilistic models, several studies explain uncertainty estimates [41, 42] and their effects [43].

**Causal Inference for Explanation.** In parallel to XAI, causal inference attempts to understand the world by identifying causal relationships from data. The popular potential outcomes framework [44–46] and causal graphical models [47] typically require either some control of the experiments (like randomized trials), or causal assumptions such as unconfoundedness. These methods can lead to strong scientific (causal) discovery, but require more care to be used correctly. In contrast, most XAI methods can be deployed to any accessible predictive models. Recently, several works have considered causal feature relevance [36], and causal contributions [48], bringing causal inference and explainability closer together [49]. However, these causal explanation methods are still centered around causal interpretations of a trained model rather than inferring the true data generating distribution.

**Uncertainty Quantification.** There are many ways to quantify uncertainty in prediction tasks, e.g., by predicting uncertainty sets or full probability distributions. A few popular methods include Gaussian Processes [50, 51], which predict full probability distributions, and quantile regression [52–54], which can give prediction intervals by minimizing the pinball loss. Conformal

prediction is a post-hoc process that can construct valid uncertainty sets or predictive distributions from heuristic notions of uncertainty [55, 56].

**Uncertainty in Explanations**   Several existing works consider uncertainty in model explanations. [57] develop Bayesian methods for quantifying uncertainty about the explanations of a single trained model. [31] give methods to compute the set of plausible variable importances for a restricted class of models. Our work differs from this work in that we aim to construct general purpose methods to quantify uncertainty about the explanations for the data generating distribution.

## 3   Framework

### 3.1   Preliminaries

We consider the task of using features $x \in \mathcal{X}$ to predict an outcome $y \in \mathcal{Y}$. Given a dataset of $n$ i.i.d. pairs $D = \{(x_1, y_1), \ldots, (x_n, y_n)\}$, the learning task is to select a probabilistic model $f$ from a model class $\mathcal{F} = \{f : \mathcal{X} \to \mathcal{P}(\mathcal{Y})\}$ that approximates the conditional distribution for $y$ given $x$. Here, $\mathcal{P}(\mathcal{Y})$ is the set of probability measures over $\mathcal{Y}$. Given a loss function $\ell(f(x), y)$, we aim to minimize the expected loss $\mathcal{L}(f) = \mathbb{E}\left[\ell(f(x), y)\right]$.

We assume the model is well-specified, so there exists some model $f^* \in \mathcal{F}$ in the model class that gives the true conditional distribution $p(y \mid x)$. We refer to $f^*$ as the *true model* since it exactly reflects the data generating distribution. Furthermore, we assume that the loss is a proper scoring rule, so $\mathcal{L}(f^*) \leq \mathcal{L}(f)$ for all other $f \in \mathcal{F}$.

Since we do not know the true model, we use some model-fitting algorithm $\mathcal{A} : \mathcal{D} \to \mathcal{F}$, where $\mathcal{D} = (\mathcal{X} \times \mathcal{Y})^n$, that takes as input a dataset $D$ and outputs a model $\hat{f} = \mathcal{A}(D)$. For example, in empirical risk minimization we choose the model $\hat{f}$ that minimizes the loss on the training data.

### 3.2   Model Explanations

We are interested in an explanation $\phi : \mathcal{F} \to \Phi$ that assigns to every model an interpretation in some space $\Phi$. The explanation $\phi$ can be a simple function of $\mathcal{F}$, such as the predicted conditional mean for a single input $x$, $\phi_x^{\mathrm{mean}}(f) = \mathbb{E}_{y \sim f(x)}\left[y\right]$, or a more complex function. For example, we can let $\phi$ map any $f \in \mathcal{F}$ to $\phi_{i,x}^{\mathrm{shap}}(f)$, to the Shapley value of the $i$-th feature applied to the feature vector $x$, with $D$ as the reference dataset; or to the average absolute Shapley value of the $i$-th feature $\phi_i^{\mathrm{shap}}(f) := \mathbb{E}_{x \sim D}[|\phi_{i,x}^{\mathrm{shap}}(f)|]$.

In binary classification where $y \in \{0, 1\}$, one can consider a counterfactual explanation $\phi_{x,+}^{\mathrm{CF}}(\hat{f})$ that returns the closest point $x'$ to $x$ such that the label is predicted to be most likely of the positive class $\hat{P}(Y = 1 \mid X = x') > 0.5$.

We are interested in the explanation of the true model $\phi(f^*)$. When $\phi$ is a simple explanation such as the conditional mean $\phi_x^{\mathrm{mean}}$, we may be able to directly estimate $\phi(f^*)$ using standard statistical techniques. When $\phi(f^*)$ is difficult to estimate directly (e.g., the Shapley values of the true model), we can first estimate the true model $f^*$ then apply the explanation $\phi$.

### 3.3   Quantifiying Uncertainty for Explanations

However, the explanation of our trained model $\phi(\hat{f})$ could be meaningfully different than the true explanation $\phi(f^*)$. For example, consider the conditional mean explanation $\phi_x^{\mathrm{mean}}(\hat{f})$ for some rare input $x$ taken from our dataset. An expressive model class could vary $\hat{f}(x)$ drastically without changing any other predictions on the dataset (and therefore only minimally change the loss). Thus, it is not enough to simply report $\phi(\hat{f})$; we instead need to quantify our uncertainty about $\phi(f^*)$.

In this work we produce *uncertainty sets* for the explanation of the true model. Using the data $D$, we construct an uncertainty set $C = C(D)$ that is guaranteed to include the true explanation with high probability

$$\mathbb{P}\left(\phi(f^*) \in C\right) \geq 1 - \alpha, \tag{1}$$

| Method | Requires Prior | Requires Posterior | Guarantee |
|--------|----------------|---------------------|-----------|
| Frequentist | No | No | $\mathbb{P}\left(\phi(f^*) \in C\right) \geq 1 - \alpha$ |
| Conformal | Yes | No | $\mathbb{P}\left(\phi(f) \in C\right) \geq 1 - \alpha$ |
| Bayesian | Yes | Yes | $\mathbb{P}\left(\phi(f) \in C \mid D\right) \geq 1 - \alpha$ |

Table 1: A comparison of the three methods we propose. The frequentist approach assumes there exists a fixed, true model $f^*$ and gives a confidence interval that includes the explanation for the true model with high probability. The randomness in the frequentist guarantee comes exclusively from the dependence of the confidence interval on the data. In the conformal and Bayesian approaches, we treat the model as a random variable $f$ distributed according to some prior distribution. Thus, the randomness in the guarantees for the conformal and Bayesian approaches is over the model $f$, the data $D$, and additional simulated randomness in $C$ we use to obtain the exact guarantee.

for some desired confidence level $1 - \alpha$ with $\alpha \in (0, 1)$. In Equation (1), the uncertainty set $C$ is random due to its dependence on the data $D$. In Section 5, we consider Bayesian models, where the model itself is a random variable. By convention, in the Bayesian perspective we will denote $f^*$ by $f$ instead to indicate that the data generating distribution is random.

From the Bayesian perspective, with an additional assumption that the posterior can be sampled exactly (see Section 5.1), one can achieve the following guarantee by employing credible intervals:

$$\mathbb{P}\left(\phi(f) \in C \mid D\right) \geq 1 - \alpha \tag{2}$$

However if one does not have access to exact samples from the posterior (Section 5.2), we propose an algorithm inspired by conformal prediction to recover a weaker guarantee, $\mathbb{P}\left(\phi(f) \in C\right) \geq 1 - \alpha$, where we no longer condition on the data. This gives a coverage guarantee over the prior rather than the posterior, unlike a typical Bayesian result.

Finally, we compare all three methods in Section 6. In our analysis we focus on two metrics: how often the uncertainty set includes the true explanation (the "coverage"), and the size of the uncertainty set. In general, higher coverage and tighter uncertainty sets are preferable.

## 4 Frequentist Explanation Interval

In this section, we introduce a method for constructing valid confidence intervals in a frequentist setting when the model class is *sufficiently simple*. We measure simplicity in a learning-theoretic sense; our results hold for model classes that satisfy uniform convergence.

Uniform convergence states that the empirical loss $\mathcal{L}_n(f) = \frac{1}{n} \sum_{i=1}^n \ell(f(x_i), y_i)$ converges to the population loss $\mathcal{L}(f) = \mathbb{E}\left[\ell(f(x), y)\right]$ "uniformly" across the model class as the number of training samples $n$ goes to infinity.

**Definition 1.** *A model class $\mathcal{F}$ has the* uniform convergence *property if, for any distributions $P$ over $\mathcal{X} \times \mathcal{Y}$, any error rate $\alpha > 0$, and any tolerance $\epsilon > 0$, there exists a sample size $n < \infty$ such that*

$$\mathbb{P}_{D \sim P}\left(\sup_{f \in \mathcal{F}} |\mathcal{L}(f) - \mathcal{L}_n(f)| \leq \epsilon\right) \geq 1 - \alpha. \tag{3}$$

We say that $\mathcal{F}$ satisfies $(\alpha, \epsilon_n)$-uniform convergence if $n$ is a sufficiently large sample size to achieve the inequality in Equation (3) with $\alpha$ and $\epsilon = \epsilon_n$. See examples of well-known uniform convergence results in Appendix **??**. First, we will note that uniform convergence gives us a confidence set for the true model. Then, we will bound the explanation of the true model by computing the most extreme explanations within this confidence set.

Uniform convergence immediately gives a bound for the excess empirical loss of the true model.

**Lemma 1.** *If $\mathcal{F}$ satisfies $(\alpha, \epsilon_n)$-uniform convergence, then with probability at least $1 - \alpha$,*

$$\mathcal{L}_n(f^*) \leq \inf_{f \in \mathcal{F}} \mathcal{L}_n(f) + 2\epsilon_n. \tag{4}$$

See Appendix A for a simple proof of Lemma 1. This bound gives us a confidence set for the true model,

$$\mathcal{F}_\alpha = \left\{ f \in \mathcal{F} : \mathcal{L}_n(f) \leq \inf_{f' \in \mathcal{F}} \mathcal{L}_n(f') + 2\epsilon_n \right\}, \tag{5}$$

which includes the true model with probability at least $1 - \alpha$. Thus, the set of explanations corresponding to $\mathcal{F}_\alpha$, namely $C_{\text{freq}} = \{\phi(f) : f \in \mathcal{F}_\alpha\}$, satisfies Equation (1).

**Proposition 1.** *Suppose $\mathcal{F}$ is well-specified and satisfies $(\alpha, \epsilon_n)$-uniform convergence. Then the confidence interval $C_{freq} = \{\phi(f) : f \in \mathcal{F}_\alpha\}$ includes the true explanation with probability at least $1 - \alpha$.*

$$\mathbb{P}\left(\phi(f^*) \in C_{freq}\right) \geq 1 - \alpha \tag{6}$$

The randomness in Equation (6) is over the dataset used to compute $C_{\text{freq}}$. Proposition 1 provides a very general guarantee that applies to all possible $f^* \in \mathcal{F}$ and any data distribution. This generality typically comes at the cost of larger confidence sets. Computing $C_{\text{freq}}$ exactly can be difficult in practice, depending on the model class. In the following Section 4.1, we elaborate on this challenge and propose an algorithm for efficiently approximating $C_{\text{freq}}$.

## 4.1 Computing Confidence Set via Pareto Frontier

For simplicity, we now consider a real-valued explanation, so $\Phi = \mathbb{R}$. However, the methods described in this section can easily be extended to vector-valued explanations, e.g., by constructing a confidence interval for each component of the vector and then applying a union bound.

Note that $C_{\text{freq}}$ is a subset of the interval $\left[\inf_{f \in \mathcal{F}_\alpha} \phi(f), \sup_{f \in \mathcal{F}_\alpha} \phi(f)\right]$. In fact, if $\mathcal{F}_\alpha$ is connected and $\phi$ is continuous, then the sets are equal up to measure 0. In turn, estimating the endpoints of this interval amounts to solving non-convex optimization problems, which can be difficult to solve exactly:

$$\begin{aligned} \text{minimize} \quad & \phi(f) \quad \text{s.t.} \quad f \in \mathcal{F}_\alpha \tag{7} \\ \text{maximize} \quad & \phi(f) \quad \text{s.t.} \quad f \in \mathcal{F}_\alpha \tag{8} \end{aligned}$$

However, we can solve a set of related unconstrained problems to approximate the solution. For Equation (7), we can define a mixed training objective:

$$J_\lambda(f) = \lambda\phi(f) + (1 - \lambda)\mathcal{L}_n(f) \tag{9}$$

By optimizing this objective for a sequence of $\lambda \in [0, 1]$, we can estimate the Pareto frontier of $\phi(f)$ and $\mathcal{L}_n(f)$ (and of $-\phi(f)$ and $\mathcal{L}_n(f)$, by flipping a sign). By choosing the first point on this Pareto frontier that satisfies the constraint, we can estimate the solution to the optimization problems posed in Equations (7) and (8). While not exact, these solutions provides an upper bound to the solution for Equation (7) and a lower bound for the solution to Equation (8).

---

**Algorithm 1:** UCEI: Uniform Convergence Explanation Intervals

**Input** : dataset $D$, mixture weights $0 \leq \lambda_1 < \cdots < \lambda_K \leq 1$

1 Estimate the ERM $\hat{f} = \min_{f \in \mathcal{F}} \mathcal{L}_n(f)$ and its empirical risk $\mathcal{L}_n(\hat{f})$

2 **for** $\lambda \in \{\lambda_1, \ldots, \lambda_K\}$ **do**

3     Optimize the mixed objective $\hat{f}_\lambda^- = \arg\min_{f \in \mathcal{F}} \lambda\phi(f) + (1 - \lambda)\mathcal{L}_n(f)$

4     Optimize the mixed objective $\hat{f}_\lambda^+ = \arg\min_{f \in \mathcal{F}} -\lambda\phi(f) + (1 - \lambda)\mathcal{L}_n(f)$

5 **end**

**Return:**

6 The confidence interval $\hat{C}_{\text{freq}}$ with lower bound

7     $\min\{\phi(\hat{f}_\lambda^-) : \mathcal{L}_n(\hat{f}_\lambda^-) \leq \mathcal{L}_n(\hat{f}) + 2\epsilon_n\}$

8 and upper bound

9     $\max\{\phi(\hat{f}_\lambda^+) : \mathcal{L}_n(\hat{f}_\lambda^+) \leq \mathcal{L}_n(\hat{f}) + 2\epsilon_n\}$

---

The frequentist guarantee relies on our ability to exactly solve the optimization problems in Equations (7) and (8). When $\phi$ is differentiable, as is the case for popular methods like LIME [9] and SHAP [8], we can use backpropagation to optimize the mixed training objective. One can also optimize each mixed objective $J_\lambda$ in parallel, or adaptively search for a $\lambda$ that gives a model with empirical loss close to $\min_{f \in \mathcal{F}} \mathcal{L}_n(f) + 2\epsilon_n$.

# 5 Bayesian Explanation Sets

The algorithm in the previous section guarantees coverage for any true function. A natural question to ask is, "can we get tighter uncertainty sets if we instead require coverage *on average*, when the true model is distributed according to some known distribution?" Consider a Bayesian model, where instead of estimating a fixed but unknown true function $f^*$, we assume the model follows a prior distribution $p(f)$. We are then interested in the posterior distribution $p(f \mid D)$, which represents our updated beliefs about the model after observing the data. A *credible set* for the posterior distribution is any subset of $\mathcal{F}$ that has probability at least $1 - \alpha$ under the posterior. We can similarly define a credible set for the explanation $\phi(f)$ as any subset of $\Phi$ that includes the explanation of a model drawn from the posterior with probability at least $1 - \alpha$.

## 5.1 Bayesian Models with a Posterior Sampler

First, we consider the case where we have sample access to the posterior distribution, i.e., a sample $f_t$ can be drawn from exactly $p(f \mid D)$ (without approximation). Credible intervals give us a natural notion of uncertainty quantification for explanations of Bayesian models; we want a set of explanations $C_{\text{Bayes}}$ that satisfies the following inequality:

$$\mathbb{P}\left(\phi(f) \in C_{\text{Bayes}} \mid D\right) \geq 1 - \alpha \tag{10}$$

Below, we describe Algorithm 2, which outputs $C_{\text{Bayes}}$ achieving the guarantee in Equation (10). Suppose that we have $T$ models $f_1, \ldots, f_T$ sampled independently from the posterior distribution $p(f \mid D)$. We can explain each model to get $T$ explanations $\phi(f_1), \ldots, \phi(f_T)$, which are independently distributed according to the posterior for the explanation $p(\phi(f) \mid D)$. We can then use these samples the estimate the quantiles of $p(\phi(f) \mid D)$. The quantiles of $p(\phi(f) \mid D)$ tell us how to construct credible intervals for the explanation. For example, the interval between the $0.05$ and $0.95$ quantiles of $p(\phi(f) \mid D)$ represents a credible interval with 90% probability under the posterior distribution. We cannot infer the quantiles of $p(\phi(f) \mid D)$ exactly from $T$ samples, but we can estimate the quantiles in such a way as to guarantee Equation (10) holds with a finite number of samples $T$, and not only asymptotically. To see this, consider drawing one more model from the posterior $f_{T+1} \sim p(f \mid D)$. Then $\phi(f_1), \ldots, \phi(f_T), \phi(f_{T+1})$ are i.i.d. explanations. It follows that $\phi(f_{T+1})$ is equally likely to be the smallest, second smallest, ..., largest element of this collection. If we define the ranking function $R(u) = \sum_{t=1}^{T} \mathbb{1}\{u \leq \phi(f_t)\}$ then $R(\phi(f_{T+1}))$ is distributed uniformly on the set $\{0, 1, 2, \ldots, T\}$. Thus, if we define the interval $C_{\text{Bayes}}$ with lower bound and upper bound as the $\lfloor \frac{\alpha}{2}(T+1) \rfloor / T$-quantile and $\lceil \left(1 - \frac{\alpha}{2}\right)(T+1) \rceil / T$-quantile (respectively) of the set $\{\phi(f_1), \ldots, \phi(f_T)\}$, then Equation (10) is guaranteed to hold, even in the finite-data regime. This is because $C_{\text{Bayes}}$ is random, even conditioned on the data $D$, since $C_{\text{Bayes}}$ also depends on the $T$ randomly drawn models from the posterior.

---

**Algorithm 2:** BEI: Bayesian Explanation Intervals

**Input** : Sampler of posterior distribution $p(f \mid D)$, explanation algorithm $\phi$, the number of samples $T$

1 **for** $t = 1, \ldots, T$ **do**
2     Sample a model $f_t \sim p(f \mid D)$
3     Compute an explanation $\phi(f_t)$ for the sampled model
4 **end**
  **Return:**
5 the confidence interval $C_{\text{Bayes}}$ with lower bound
6 Quantile($\{\phi(f_1), \ldots, \phi(f_T)\}; \lfloor \frac{\alpha}{2}(T+1) \rfloor / T$)
7     and upper bound
8 Quantile($\{\phi(f_1), \ldots, \phi(f_T)\}; \lceil \left(1 - \frac{\alpha}{2}\right)(T+1) \rceil / T$)

---

## 5.2 Bayesian Models with a Prior Sampler

In the previous section, we showed that one can get exact uncertainty sets for Bayesian models if an exact posteior sampler is available. However, for many Bayesian models, such as Bayesian neural networks [50, chap 5.7] and latent Dirichlet allocation [58], it would be prohibitively expensive to sample from the exact posterior distribution. In such settings, practitioners often resort to approximating the posterior distribution, e.g., by using variational inference or Markov Chain Monte Carlo

samplers [50, chap 10,11]. However, when we only have access to samples from an approximate posterior distribution, it is not obvious how we can salvage our exact credible interval guarantee in Equation (10). In this section, we provide an algorithm that works without exact posterior samples, at the cost of providing weaker guarantees. Specifically, we guarantee validity with respect to the *prior* instead of the posterior. To do this, we recruit tools from conformal inference (See Section 2.)

Conformal inference is most often applied in frequentist settings, and allows one to construct prediction sets $C(x_i)$ for each new label $y_i$ that enjoy finite-sample coverage guarantees. Specifically, the guarantee is that $\mathbb{P}\left(y \in C(\hat{f}(x))\right) \geq 1 - \alpha$, where $x$ and $y$ are random, and $C$ is a random function of $\hat{f}(x)$ that also depends on held out calibration samples. Here, $\hat{f}$ is an arbitrary predictor for $y$ that takes $x$ as input. (Technically, there is an assumption that $\hat{f}$ treats the data symmetrically, but this is not important for our discussion here.) Conformal prediction requires $T$ calibration samples for which both the prediction $\hat{f}(x)$ and the outcome $y$ are observed.

We give a strategy for computing an uncertainty set $C_{\text{conformal}}$ that is analogous to the conformal inference result, except that instead of giving an uncertainty set for a new label, we give an uncertainty set for the explanation of a model. The central challenge to applying conformal inference to our setting is obtaining our calibration samples; we usually do not know the true model explanation for any dataset. We get around this problem by sampling models i.i.d. from our prior distribution, $f_1, \ldots, f_t, \ldots, f_T \sim p(f)$. Recall that since our models are probabilistic, given an input $x_i$, we can sample a label $y_i^t \sim f_t(x_i)$ from the distribution predicted by the model. By pairing each original input $x_i$ with the corresponding resampled label $y_i^t$, we have a dataset $D_t = \{(x_1, y_1^t), \ldots, (x_n, y_n^t)\}$ drawn from the model $f_t$. We can then train a model $\hat{f}_t = \mathcal{A}(D_t)$ on this new dataset. This gives us $T$ examples where we can observe the ground truth explanation $\phi(f_t)$ and an estimated explanation $\phi(\hat{f}_t)$. We compare how close $\phi(f_t)$ and $\phi(\hat{f}_t)$ tend to be using a *nonconformity score*, such as the distance $\|\phi(f_t) - \phi(\hat{f}_t)\|$. These examples act as our calibration dataset in Algorithm 3.

---

**Algorithm 3:** CEI: Conformal Explanation Intervals

---

**Input** : Model-fitting algorithm $\mathcal{A}$, dataset $D = (x_1, y_1), \ldots, (x_n, y_n)$
**Input** : Nonconformity score $s : \Phi \times \Phi \to \mathbb{R}$
1 Train a model $\hat{f} = \mathcal{A}(D)$ using the dataset
2 Explain the trained model $\hat{\phi} = \phi(\hat{f})$
3 **for** $t = 1, \ldots, T$ **do**
4 $\quad$ Sample a model $f_t \sim p(f)$
5 $\quad$ Sample a dataset of labels $y_i^t \sim f_t(x_i)$
6 $\quad$ Define the synthetic dataset $D_t = \{(x_1, y_i^t), \ldots, (x_n, y_n^t)\}$
7 $\quad$ Train a model $\hat{f}_t = \mathcal{A}(D_t)$
8 $\quad$ Explain the sampled model $\phi_t = \phi(f_t)$ and the trained model $\hat{\phi}_t = \phi(\hat{f}_t)$
9 $\quad$ Compute the nonconformity score $s_t = s(\phi_t, \hat{\phi}_t)$
10 **end**
11 Set the threshold $\tau$ as the $\lceil(1 - \alpha)(T + 1)\rceil/T$-quantile of the set $\{s_1, \ldots, s_T\}$
**Return:** $C_{\text{conformal}} = \{\varphi \in \Phi : s(\varphi, \phi(\hat{f})) \leq \tau\}$

---

The uncertainty set $C_{\text{conformal}}$ has the following guarantee:

**Proposition 2.** *The confidence interval $C_{conformal}$ given by Algorithm 3 includes the model $f$ with high probability over the prior distribution:*

$$\mathbb{P}_{f \sim p(f)}\left(\phi(f) \in C_{conformal}\right) \geq 1 - \alpha \tag{11}$$

Here, $f$ is random due to the prior and $C_{\text{conformal}}$ is random due to the data and the calibration samples. Note that computing $C_{\text{conformal}}$ does not rely on us knowing the posterior distribution. We only need some algorithm $\mathcal{A}$, such as an empirical risk minimizer that, given a dataset, returns a model $\hat{f} \in \mathcal{F}$. However, this guarantee is weaker than the guarantee we got when we had access to the posterior distribution in Equation (10). Note that we are not conditioning on the data in Equation (11), and so $C_{\text{conformal}}$ is not necessarily a credible interval under the posterior. Furthermore, by

|             | Coverage            | Interval Width      |
|-------------|---------------------|---------------------|
| Naive       | $0.5067 \pm 0.002$  | $0.2522 \pm 0.001$  |
| Frequentist | $1.0000 \pm 0.000$  | $3.9310 \pm 0.003$  |
| Bayesian    | $0.9500 \pm 0.001$  | $0.9679 \pm 0.001$  |
| Conformal   | $0.9600 \pm 0.001$  | $1.0493 \pm 0.001$  |

Table 2: A comparison of the coverage rate and interval width of the uncertainty sets from each method on synthetic data. Each of our proposed methods achieves the desired coverage of 0.95. The frequentist method is overly cautious due to its reliance on conservative learning theory results.

integrating over the dataset $D$, the (conditioned) guarantee in Equation (10) can be seen to imply the (unconditioned) guarantee in Equation (11). The condition in Proposition 2 is also satisfied by any credible interval $C$ for the prior distribution. However, one should typically find that $C_{\text{conformal}}$ can give much tighter uncertainty sets than this naive strategy since it is adaptive to the data. In this way, $C_{\text{conformal}}$ can be viewed as an intermediate solution between a credible set for the prior and a credible set for the posterior; it is more adaptive to the data than the former but not as adaptive as the latter.

## 6 Experiments

We perform experiments on synthetic and real-world datasets. Synthetic datasets allow us to generate uncertainty sets for explanations where we know the true data generating distribution. This means that we can validate the coverage rates of our methods, which is difficult to do with real-world data where we do not know the data generating distribution. Experiments with real-world datasets give insight on how our methods scale to larger datasets and realistic distributions.

### 6.1 Experimental Setup

We perform three experiments under the Shapley value explainer. In a synthetic experiment, we first validate that the frequentist, Bayesian, and conformal methods all achieve the desired coverage rate and compare the size of the uncertainty sets each method gives. In the second synthetic experiment, we test the robustness of the Bayesian and conformal methods to violations of their assumptions by exploring settings where the prior distribution is misspecified. Lastly, we apply the conformal method to infer feature importance scores for a variety of real-world datasets.

**Testing Coverage on Synthetic Data**  We consider a regression problem with three features. The data is generated according to the following distributions:

$$x_i \sim \mathcal{N} \left( \mu = \mathbf{0}, \Sigma = \begin{bmatrix} 1 & 0.99 & 0 \\ 0.99 & 1 & 0 \\ 0 & 0 & 1 \end{bmatrix} \right)$$

$$y_i = \theta^\top x + \epsilon_i, \qquad \epsilon_i \sim \mathcal{N}(0, 1)$$

We fit linear models $f_\beta(x) = \mathcal{N}(\cdot \; ; \mu = \beta^\top x, \sigma^2 = 1)$ that predict Gaussian distributions for $y$, where $\beta \in \mathbb{R}^3$. We independently sample $M = 100$ true models $\theta_1, \ldots, \theta_M \sim \mathcal{N}(\mathbf{0}, I_3)$, and for each true model $\theta_m$, we sample a dataset $D_\theta$ consisting $n = 100$ examples. For the Bayesian and conformal methods, we the correct prior $\mathcal{N}(0, I_3)$ for $\theta$. For the naive and conformal methods, we fit $T = 100$ linear models using Ridge regression. For the frequentist method, we use a standard uniform convergence result for linear regression. We use closed-form expression of Shapley value under the linear case. Refer to Appendix B for the detailed Rademacher complexity chosen and Shapley expression. For each method, we record the portion of the time that the uncertainty set includes the true explanation: $\text{coverage}(C) = \frac{1}{3M} \sum_{m=1}^{M} \sum_{i=1}^{n} \mathbb{1} \left\{ \phi(\theta_m^{(i)}) \in C^{(i)} \right\}$. For each method, the targeted coverage rate is 0.95. We also report the average width of the uncertainty set for each method.

**Testing Coverage under Model Misspecification**  We also explore how the Bayesian and conformal methods perform when the prior distribution is incorrect. We re-run the coverage experiment

|           |                | Coverage          | Interval Width    |
|-----------|----------------|-------------------|-------------------|
| Bayesian  | well-specified | $0.9500 \pm 0.001$ | $0.9679 \pm 0.001$ |
|           | wrong mean     | $0.7511 \pm 0.001$ | $0.8620 \pm 0.001$ |
|           | wrong var      | $0.9600 \pm 0.001$ | $0.8720 \pm 0.001$ |
| Conformal | well-specified | $0.9600 \pm 0.001$ | $1.0493 \pm 0.001$ |
|           | wrong mean     | $0.9867 \pm 0.001$ | $1.0634 \pm 0.001$ |
|           | wrong var      | $0.9800 \pm 0.001$ | $1.0578 \pm 0.001$ |

Table 3: A comparison of the Bayesian and conformal methods when their assumptions are not met, due to a misspecified prior distribution. The Bayesian method loses coverage when the mean is misspecified in this case. The conformal method becomes unnecessarily cautious, giving wider intervals than necessary.

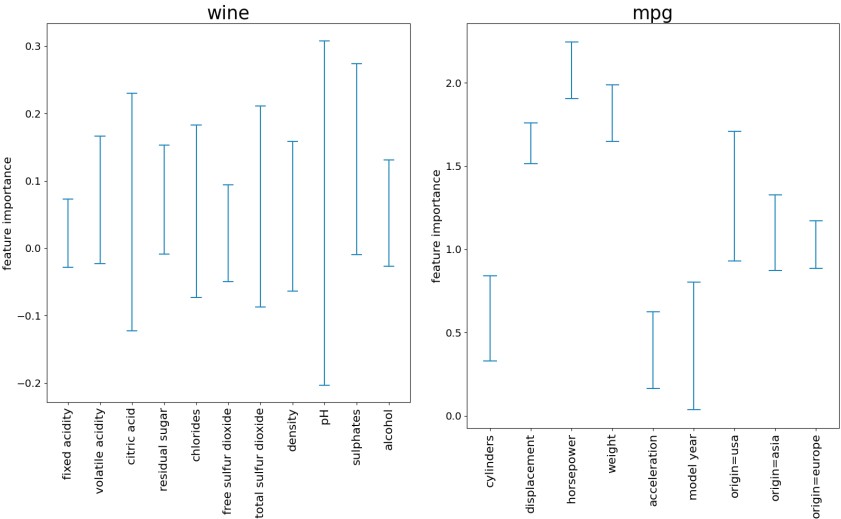

Figure 3: Feature importance scores (as measured by the mean Shapley value of each feature across the dataset) computed using the conformal method. For some datasets (e.g., MPG) there are significant differences between the importances assigned to different features. For other datasets (e.g., WINE), conclusions are more difficult to make. When features have overlapping feature importance uncertainty sets, it indicates that practitioners should be cautious when drawing conclusions.

for the Bayesian and conformal methods, except with the true models sampled with a different mean $\mathcal{N}(\mathbf{1}, I_3)$ and with a different variance $\mathcal{N}\left(\mathbf{0}, \frac{1}{2}I_3\right)$. All other aspects of the experiment are unchanged.

**Real-world Data Experiments** We consider eight tabular regression datasets: (all results except for WINE and MPG deferred to Appendix D). In each case, we train a neural network to predict a real-valued label. The model outputs a mean and variance for a Gaussian distribution, and is trained with the negative log-likelihood loss. The architecture has 2 hidden layers, each with 100 neurons, and uses ReLU activations. We compute uncertainty sets for the explanation of the true model using the conformal explanation intervals method. For the explanation, we use the average of the absolute value of the Shapley value of the feature across the dataset (a measure of feature importance). We set the prior distribution for each weight to be Gaussian with zero mean and variance as the reciprocal of the dimension of the layer. The prior for the biases are standard Gaussian distributions. We generate $T = 100$ calibration examples by sampling models from the prior.

## 6.2 Experimental Results

**Testing Coverage on Synthetic Data** We find that each of our proposed method achieves a coverage of at least 95% (the naive method has coverage close to $50\%$). See Table 2 for the complete results. The frequentist coverage interval tends to be overly conservative due to the worst-case per-

spective of uniform convergence, leading to a roughly 4x greater interval width when compared to the other methods, and 100% coverage in our experiments.

**Real-world Data Experiments**  We find that the strength of conclusions that can be drawn from the conformal method varies across datasets. For example, for the MPG dataset, the features `displacement`, `horsepower`, and `weight` have high importance with low uncertainty. However, in the PROTEIN dataset, it is difficult to make any meaningful conclusions about the relative importance of features, possibly due to the existence of competing models that use different features. Seven additional experimental results are included in Appendix D.

## 7   Discussion

We offer guidance to a practitioner deciding between the frequentist, Bayesian, and conformal approaches. If we do not have a prior for our model, the frequentist approach can give strong guarantees at the cost of large uncertainty sets. For Bayesian models, we recommend using the fully Bayesian approach when the posterior admits exact sampling, since this gives stronger guarantees. When an exact posterior sampler is not available, the conformal approach can recover a weaker guarantee that can still give tight uncertainty sets.

In this work we show how to give confidence sets for explanations of the data generating process. We caution, however, that explanations of the true data generating distribution do not in general have causal implications. Still, we hope that formalizing a connection between model explanations and the data generating distribution can help users understand which explanations are the result of spurious correlations, and which are meaningful.

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

## A  Proofs

**Lemma 1.** *If uniform convergence holds, then with probability at least $1 - \alpha$,*

$$\mathcal{L}_n(f^*) \leq \inf_{f \in \mathcal{F}} \mathcal{L}_n(f) + 2\epsilon_n.$$

*Proof of Lemma 1.* To see this, denote by the event $E = \{\sup_{f \in \mathcal{F}} |\mathcal{L}(f) - \mathcal{L}_n(f)| \leq \epsilon\}$, which occurs with probability at least $1 - \alpha$. If the event $E$ occurs, then for all models $f \in \mathcal{F}$ we have

$$
\begin{aligned}
\mathcal{L}_n(f^*) &\leq \mathcal{L}(f^*) + \epsilon_n & \text{($E$ occurred)} \\
&\leq \mathcal{L}(f) + \epsilon_n & \text{(Optimality of $f^*$)} \\
&\leq \mathcal{L}_n(f) + 2\epsilon_n. & \text{($E$ occurred)}
\end{aligned}
$$

Since this inequality holds for all $f \in \mathcal{F}$, it also holds for the infimum over $\mathcal{F}$. This gives the result in Equation (4). $\square$

## B  Example: Linear Regression

We will walk through an example in which we infer the Shapley values of an unknown linear function.

### B.1  Shapley Value Derivation for Linear Models

Consider a linear model given by $f_\theta(x) = \theta^\top x$, where $\theta \in \mathbb{R}^d$ and $x \in \mathbb{R}^d$. We use the capital $X$ to represent the random variable for the feature vector, and the lower case $x$ to represent a fixed value for this random variable. Given a coalition of features $S \subseteq \{1, \ldots, d\}$, we can write the feature vector $x$ as $x = [x_S, x_{\overline{S}}]$ where $x_S = \{x_i : i \in S\}$ and $x_{\overline{S}} = \{x_i : i \notin S\}$. The prediction associated with the coalition $x_S$ can be defined as

$$
\begin{aligned}
f_\theta(x_S) &:= \mathbb{E}\left[f_\theta([x_S, X_{\overline{S}}])\right] & (12) \\
&= \mathbb{E}\left[\theta^\top [x_S, X_{\overline{S}}]\right] & (13) \\
&= \theta_S^\top x_S + \theta_{\overline{S}}^\top \mathbb{E}\left[X_{\overline{S}}\right] & (14) \\
&= \theta_S^\top x_S + \theta_{\overline{S}}^\top x_{\overline{S}} + \theta_{\overline{S}}^\top \mathbb{E}\left[X_{\overline{S}} - x_{\overline{S}}\right] & (15) \\
&= f_\theta(x) + \sum_{i \notin S} \theta_i \mathbb{E}\left[X_i - x_i\right] & (16)
\end{aligned}
$$

The Shapley value for a linear model on a particular instance $x$ can then be written as:

$$
\phi_i^x(f_\theta) = \sum_{S \subseteq [d] \setminus i} \frac{|S|!(d - |S| - 1)!}{d!}(f_\theta(x_{S \cup \{i\}}) - f_\theta(x_S)) \tag{17}
$$

$$
= \sum_{S \subseteq [d] \setminus i} \frac{|S|!(d - |S| - 1)!}{d!}\left(\left(f_\theta(x) + \sum_{j \notin S \cup \{i\}} \theta_j \mathbb{E}\left[X_j - x_j\right]\right) - \left(f_\theta(x) + \sum_{j \notin S} \theta_j \mathbb{E}\left[X_j - x_j\right]\right)\right) \tag{18}
$$

$$
= \sum_{S \subseteq [d] \setminus i} \frac{|S|!(d - |S| - 1)!}{d!}\theta_i \mathbb{E}\left[X_i - x_i\right] \tag{19}
$$

$$
= \frac{1}{Z}\theta_i \mathbb{E}\left[X_i - x_i\right] \tag{20}
$$

where $Z = \left( \sum_{S \subseteq [d] \setminus i} \frac{|S|!(d-|S|-1)!}{d!} \right)^{-1}$ is the normalizing constant. Using the efficiency property of the Shapley value which states that $\sum_{i \in [d]} \phi_i^x(f_\theta) = f_\theta(x)$, we can compute the normalizing constant $Z$ by noting that:

$$f_\theta(x) = \sum_{i \in [d]} \phi_i^x(f_\theta) \tag{21}$$

$$= \frac{1}{Z} \sum_{i \in [d]} \theta_i \mathbb{E}\left[ X_i - x_i \right] \tag{22}$$

giving us that

$$Z = \frac{1}{f_\theta(x)} \sum_{i \in [d]} \theta_i \mathbb{E}\left[ X_i - x_i \right] \tag{23}$$

We can then write the Shapley value as

$$\phi_i^x(f_\theta) = \frac{f_\theta(x)}{\theta^\top \left( \mathbb{E}\left[ X \right] - x \right)} \theta_i \mathbb{E}\left[ X_i - x_i \right] \tag{24}$$

when the denominator is nonzero, and $\phi_i^x(f_\theta) = 0$ when the denominator is equal to zero.

## B.2 Uniform Convergence for Squared Loss of Linear Models

Uniform convergence results bound (with high probability) the disagreement $\sup_{f \in \mathcal{F}} |\mathcal{L}(f) - \mathcal{L}_n(f)|$ between the sample loss and population loss over a model class. Uniform convergence results are often stated in terms of the Vapnik–Chervonenkis dimension, Rademacher complexity, Gaussian complexity, covering number, and other notions of complexity of the model class. Below, we display a few standard results that, together, give a uniform convergence result for linear models with squared error loss.

**Theorem 1** ([59]). *Let $\mathcal{F} = \left\{ x \mapsto \theta^\top x : \|\theta\|_p \leq w \right\}$ be a family of linear functions defined over $\mathbb{R}^d$ with bounded weight in $\ell_p$-norm. Then, the empirical Rademacher complexity of $\mathcal{F}$ for a sample $S = (\mathbf{x}_1, \ldots, \mathbf{x}_n)$ admits the following upper bounds:*

$$\hat{\mathcal{R}}_S\left( \mathcal{F} \right) \leq \frac{w}{n} \|\mathbf{X}\|_{\mathrm{Fr}}$$

*where $\mathbf{X}$ is the $d \times n$-matrix with $\mathbf{x}_i$s as columns: $\mathbf{X} = [\mathbf{x}_1 \ldots \mathbf{x}_n]$.*

**Theorem 2.** *For the squared error loss $\ell(y, \hat{y}) = (y - \hat{y})^2$, let $\ell \circ \mathcal{F} := \left\{ x \mapsto (f_\theta(x) - f^*(x))^2 : f_\theta \in \mathcal{F} \right\}$. Assume that $\sup_{x \in \mathcal{X}, f_\theta \in \mathcal{F}} (f_\theta(x) - f^*(x))^2 \leq M^2$. Then for any sample $S = \{x_1, \ldots, x_n\}$,*

$$\hat{\mathcal{R}}_S(\ell \circ \mathcal{F}) \leq 2M \hat{\mathcal{R}}_S(\mathcal{F}) \tag{25}$$

**Theorem 3.** *With probability at least $1 - \delta$, for all $f \in \mathcal{F}$ and distributions over $\mathcal{X} \times \mathcal{Y}$ the following holds:*

$$\mathcal{L}(f) - \mathcal{L}_n(f) \leq 2\hat{\mathcal{R}}_S(\ell \circ \mathcal{F}) + 3\sqrt{\frac{\ln 1/\delta}{2n}} \tag{26}$$

Together, Theorems 1-3 give us the following result for linear models with the squared error loss. For all $f \in \mathcal{F}$ and any distribution over $\mathcal{X} \times \mathcal{Y}$, with probability at least $1 - \delta$,

$$\mathcal{L}(f) - \mathcal{L}_n(f) \leq \frac{4Mw}{n} \|\mathbf{X}\|_{\mathrm{Fr}} + 3\sqrt{\frac{\ln 1/\delta}{2n}} \tag{27}$$

where $\mathbf{X}, M, w, n, \delta$ are defined as in Theorems 1-3.

## C    Conformal Prediction

Our Algorithm 3 is heavily inspired by conformal prediction, a simple and effective method for constructing statistically rigorous uncertainty intervals [56, 60, 61]. In the standard conformal prediction setup, we have an i.i.d. calibration dataset $(X_1, Y_1), \dots, (X_n, Y_n)$ and some new input $X_{\text{test}}$ for which we want to predict the label $Y_{\text{test}}$. Given a black-box model $\hat{f}$ trained (on separate data) to predict the label, we want to construct uncertainty estimates for the predictions made by the black-box model. Conformal prediction tells us how to construct an uncertainty interval $C_{\text{test}}$ such that the ground truth value $Y_{\text{test}}$ is included in the uncertainty interval with some chosen probability $1 - \alpha$, such at 95%:

$$P(Y_{test} \in C_{\text{test}}) \geq 1 - \alpha \tag{28}$$

Perhaps the biggest advantage of conformal prediction is that it applies under extremely weak assumptions. As long as the model-fitting algorithm $\mathcal{A}$ treats the data symmetrically (e.g., a time series forecast that weighs recent data more heavily does not treat data symmetrically) and the data is i.i.d. (in fact, the weaker condition of exchangeability is sufficient), the uncertainty interval will be valid. No additional distributional assumptions on the data generating process are needed.

The central component of a conformal prediction algorithm is the nonconformity score. The nonconformity score $s(Y, \hat{f}(X))$ evaluates the disagreement between the observed outcome and the model's prediction. For a binary classifier that outputs a probability $\hat{p}(X_{\text{test}})$ that $Y_{\text{test}} = 1$, a reasonable nonconformity score would be $s(Y_{\text{test}}, \hat{p}(X_{\text{test}})) = |Y_{\text{test}} - \hat{p}(X_{\text{test}})|$. For a regression model, a reasonable nonconformity score would be $s(Y_{\text{test}}, \hat{\mu}(X_{\text{test}})) = |Y_{\text{test}} - \hat{\mu}(X_{\text{test}})|$ where $\hat{\mu}(X_{\text{test}})$ is the predicted mean.

Importantly for our purposes, conformal prediction requires us to have i.i.d. examples to calibrate our uncertainties. When we want an uncertainty interval for an outcome $Y_{\text{test}}$, this is not a problem since we often have access to pairs of true outcomes $Y_i$ and predicted outcomes $\hat{f}(X_i)$. However, in our case we want an uncertainty interval for the explanation $\phi(f)$. Problematically, we never observe a true explanation because we never observe i.i.d. examples of a "true explanation" $\phi(f)$ and an estimated explanation $\phi(\hat{f})$. Usually, we only observe a single dataset generated from a single model $f$. However, if we have a prior distribution for the model, then we can simulate i.i.d. examples of true explanations $\phi(f)$ and estimated explanations $\phi(\hat{f})$ then run conformal prediction.

## D    Additional Experimental Results

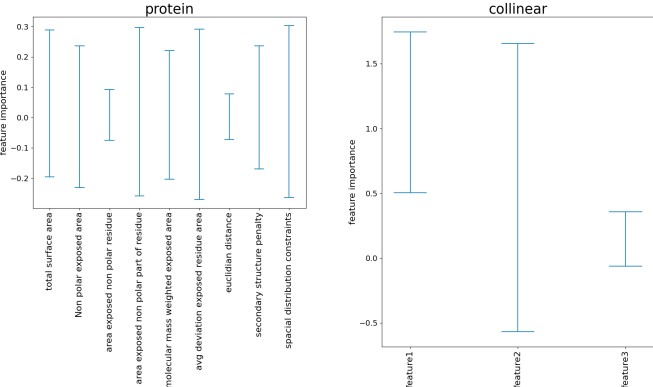

Figure 4: Feature importance scores (as measured by the mean Shapley value of each feature across the dataset). Confidence intervals are computed using the conformal explanation intervals method.

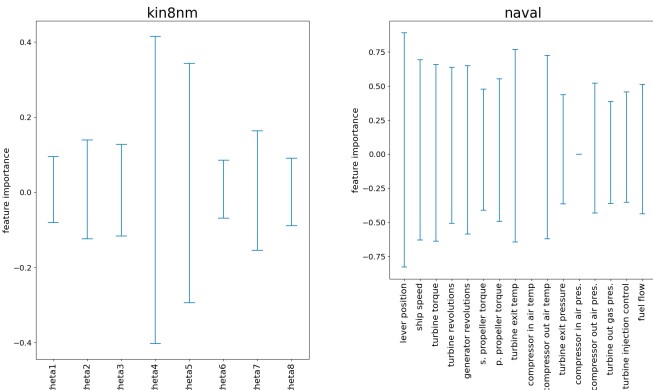

Figure 5: Feature importance scores (as measured by the mean Shapley value of each feature across the dataset). Confidence intervals are computed using the conformal explanation intervals method.

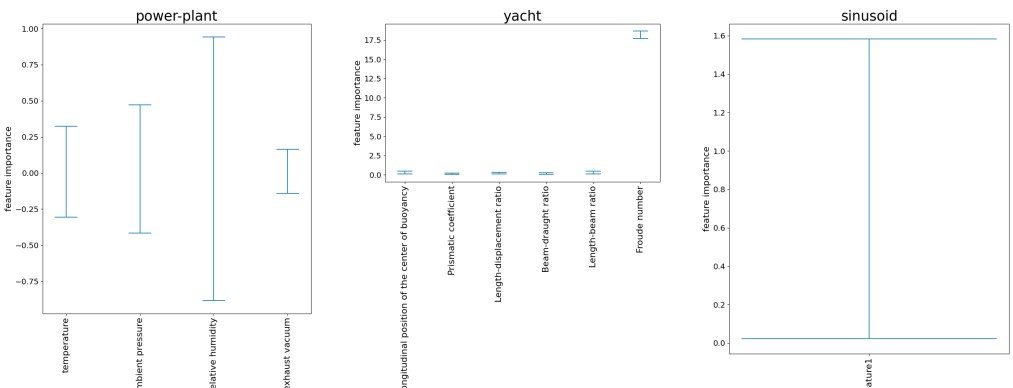

Figure 6: Feature importance scores (as measured by the mean Shapley value of each feature across the dataset). Confidence intervals are computed using the conformal explanation intervals method.

