# OpenReview forum: "But Are You Sure? Quantifying Uncertainty in Model Explanations"
_NeurIPS.cc/2022/Workshop/TSRML — TSRML2022_

### Official Review · Reviewer_y4kV · 2022-10-17
**Interesting direction that needs a bit more clarity**

**Overall Rating:** 4

**Summary:**

This paper introduces a method to quantify uncertainty over the true explanation prediction. These introduce both bayesian and frequentist treatment to that end.

**Strengths:**

- Interesting and important problem

**Weaknesses:**

The problem setup is quite hard to piece apart and requires a bit more careful attention. It seems that the goal is to find uncertainty for the ground truth model... but it what regard? It's unclear if this is a global or local explanation. Does the explanation need to explain the entire model or just a small part of it?

Also, what exactly is $\Phi$? Is it a vector of importances for each feature? I'm not exactly sure how to understand this when its also possible to be a a counterfactual explanation, which is just another data point.



**Overall Recommendation:**

I think that this paper introduces an interesting direction but it's quite challenging to follow the problem setup. The authors should consider revising the setup to make it more clear.

**Review Confidence:**

4: The reviewer is confident but not absolutely certain that the evaluation is correct

---

### Official Review · Reviewer_5JVx · 2022-10-19

**Overall Rating:** 7

**Summary:**

Existing explainability methods focus on explaining a single fitted model, however these explanations are often unstable and inconsistent across different models and therefore can not be used to explain data generating models. This paper proposes a principled way for constructing uncertainty set for the explanation of the data generating model. They provide theoretical guarantees for the method and empirically evaluate the algorithms on synthetic data in addition to conducting experiments on real-world datasets.

**Strengths:**

1. The paper considers an important problem of quantifying the uncertainty in explanations through constructing uncertainty sets for explanations.
2. The authors propose three algorithms: for tractable Bayesian models, for intractable Bayesian models, and for frequentist models.
3. For each of three algorithms authors provide theoretical guarantees that found uncertainty sets include the correct explanation.
4. The paper validates the algorithms on synthetic dataset and provide feature importance uncertainty analysis for real-world datasets.

**Weaknesses:**

I did not find any significant weaknesses in the paper. I personally would prefer more empirical experiments validating the proposed algorithms to demonstrate how the theoretical guarantees work in practice.

**Overall Recommendation:**

I recommend this paper for acceptance because it approaches an important problem of quantifying uncertainty of explanations and proposes three algorithms for different types of models. Although the proposed approaches have certain limitations (dependence on prior distribution), I believe that the work makes practical contribution to the field of explainability in machine learning.

**Review Confidence:**

3: The reviewer is fairly confident that the evaluation is correct

---

### Official Review · Reviewer_i9bF · 2022-10-20
**Interesting work but weak experimental results**

**Overall Rating:** 6

**Summary:**

The authors advocate using an uncertainty set for model explanations instead of relying on only the explanation from the best-fitting model. They propose three systematic means of constructing such an uncertainty set that contains the explanation for the true (unknown) model with high probability and show their finite-sample guarantees.

**Strengths:**

- The problem is well-motivated with a clear presentation.
- The proposed algorithms and proofs appear to be technically correct
- The proposed algorithms addressing both frequentist and Bayesian views of uncertainty quantification cover a wide range of applications and problems


**Weaknesses:**

The produced intervals in the numerical experiments do not appear to be sufficient for scientific inquiry for many datasets. While the proposed construction of an uncertainty set provably contains the true explanations with high probability, it appears that they tend to produce a very large set, thus becoming less informative and undermining their usefulness in practice in “extracting principles” relevant to scientific inquiry. In this case, how to best use the less tight uncertainty sets? Would the author suggest ways to improve the tightness of the uncertainty set? Or would it be possible to further use the quantified uncertainty to improve the explanation models?

**Overall Recommendation:**

The paper is well-written and highly relevant to the topic but has rather weak experiment results.

**Review Confidence:**

3: The reviewer is fairly confident that the evaluation is correct

---

### Decision · Program_Chairs · 2022-10-23

**Decision:**

Accept

**Comment:**

Reviewers have concerns on the clarity of this paper, and the lack of sufficient empirical results to demonstrate the proposed theoretical guarantee. I hope the comments from reviewers are helpful and the authors can improve this paper in the final version.